



# Ionospheric Anomalies Associated with Mw7.3 Iran-Iraq Border Earthquake and a Moderate Magnetic Storm

Erman ŞENTÜRK[1], Samed INYURT[2], İbrahim SERTÇELİK[3]

[1]Department of Geomatics Engineering, Kocaeli University, Turkey
[2]Department of Geomatics Engineering, Gaziosmanpaşa University, Turkey
[3]Department of Geophysical Engineering, Kocaeli University, Turkey

**Correspondence:** Erman Şentürk (erman.senturk@kocaeli.edu.tr)

**Abstract.** The analysis of the unexpected ionospheric phases before large earthquakes is a popular approach in earthquake prediction studies. In this study, the Total Electron Content (TEC) data of five International GNSS Service (IGS) stations and the Global Ionosphere Maps (GIMs) were used. The Short-time Fourier Transform (STFT) and a running median process were applied on the TEC time series to detect abnormalities before the Mw7.3 Iran-Iraq border earthquake on November 12, 2017. The analyzes showed positive anomalies 8-9 days before the earthquake and some positive/negative anomalies 1-6 days before the earthquake. These anomalies were cross-checked by space weather indices Kp, Dst, F10.7, Bz component of the interplanetary magnetic field (IMF Bz), electric field (Ey), and plasma speed ($V_{SW}$). The results showed that the anomalies 1-6 days before the earthquake caused by a moderate magnetic storm. Also, the positive anomalies 8-9 days before the earthquake should be related to the Iran-Iraq border earthquake due to quiet space weather, local dispersion, and proximity to the epicenter.

## 1 Introduction

The ionosphere is a three-dimensional dispersive atmosphere layer. The layer locates above approximately 50-1000 km from the Earth's surface and includes molecules with potential for photoionization. The molecules are separated into protons and electrons when exposed to light energy emitted from the sun. Electrons separated from molecules effect the propagation of electromagnetic signals traveling between space and earth. The degree of effect is a function of the number of free electrons. The sun is the primary determiner of the number of electrons and causes permanent and regular ionospheric trends such as daily, 27-day, seasonal, semi-annual, annual, and 11-year. The number of electrons also increase/decrease due to disturbed space-weather (Bagiya et al., 2009), earthquakes (Liu et al., 2004; Şentürk et al., 2018), tsunamis (Occhipinti et al., 2013), volcanic eruptions (Dautermann et al., 2009), hurricanes (Chou et al., 2017) and anthropogenic events (Lin et al., 2017). These events generally cause non-secular changes, which are commonly named as ionospheric disturbances/anomalies.

In recent decades, seismoionospheric studies have become quite popular. The first case was reported for Good Friday Alaska Earthquake of 1964 (Davies and Baker, 1965; Leonard and Barnes, 1965). In those years, data of ionosonde, radio waves, and topside sounding were used to analyze ionospheric anomalies before earthquakes (Gokhberg, 1983; Molchanov et al., 1992; Pulinets et al., 1998; Liu et al., 2000). Calais and Minster (1995) firstly used GPS observations for seismoionospheric analysis of the Mw 6.7 Northridge earthquake.



Global Navigation Satellite System (GNSS) technology provides low-cost, high accuracy, near real-time, and continuous ionospheric data. GNSS based TEC data is preferred in many subsequent seismoionospheric studies related to large earthquakes (Liu et al., 2004, 2010; Fuying et al., 2011; Yildirim et al., 2016; Ulukavak and Yalcinkaya, 2017; Yan et al., 2017; Ke et al., 2018; Şentürk et al., 2018; Tariq et al., 2019). Liu et al. (2004) investigated 20 earthquakes with a magnitude greater than 6 in
Taiwan between 1999 and 2002. They used the GPS based TEC data and applied the 15-days moving median and quartile range method to the TEC variation. The results showed that ionospheric abnormalities were detected before earthquakes, with an 80% success rate. Liu et al. (2010) reported seismoionospheric precursors of the 2004 M=9.1 Sumatra-Andaman Earthquake due to anomalous decreases in the TEC variation five days before the earthquake. Fuying et al. (2011) used the Kalman filter method to detect the
abnormal changes of TEC variations before and after the Wenchuan Ms8.0 earthquake. The TEC data were calculated from the GPS observations observed by the Crustal Movement Observation Network of China (CMONOC). The result showed that the Kalman filter is reasonable and reliable in detecting TEC anomalies associated with large earthquakes. Yildirim et al. (2016) utilized 4 Continuously Operating Reference Stations in Turkey (CORS-TR) and 11 IGS and EUREF Permanent Network (EPN) stations to
investigate the ionospheric disturbances related to Mw 6.5 offshore in the Aegean Sea earthquake on 24 May 2014. TEC data obtained from Precise Point Positioning and GIMs showed that the TEC values anomalously increased 2-4 TECU 3 days before the earthquake and decreased 4-5 TECU on the day before the earthquake. Ulukavak and Yalcinkaya (2017) used GNSS based TEC data of 6 IGS stations to determine the pre-earthquake ionospheric anomalies before the Mw 7.2 Baja California earthquake on 4
April 2010. The results showed both positive and negative ionospheric anomalies occurred one to five days before the earthquake. Yan et al. (2017) utilized data of CMONOC and IGS to statistically investigate the TEC anomalies before 30 Mw6.0+ earthquakes from 2000 to 2010 in China. TEC anomalies were detected before 20 earthquakes, nearly 67%. Ke et al. (2018) used a Linear Model between TEC and F10.7 (LMTF) to detect seismoionospheric TEC anomalies before and after the Nepal
earthquake 2015. The method was compared with Sliding Quartile and Kalman filter methods. They found that LMTF is more effective in detecting the TEC anomalies caused by the Nepal earthquake in temporal and spatial. Şentürk et al. (2018) comprehensively analyzed the ionospheric anomalies before the Mw7.1 Van earthquake on 23 October 2011 with temporal, spatial, and spectral methods. The results showed a 2-8 TECU increase in the TEC time series of 28 GNSS stations and GIMs before the Van earthquake on 9
October, 15-16 October, and 21-23 October. Tariq et al. (2019) used GNSS based TEC data to detect seismoionospheric anomalies of three major earthquakes (M>7.0) in Nepal and the Iran-Iraq border during 2015-2017. The ionospheric precursors of three earthquakes generally occur within ten days, about 08:00-12:00 UT in the daytime. The temporal and spatial statistical tests showed that the abnormal positive TEC changes were detected nine days before the Mw7.3 Iraq earthquake.

There is still no consensus on the physical process of the changes in the ionosphere before earthquakes, but several assumptions have been made about the subject (Toutain and Baubron, 1998; Pulinets et al., 2006; Namgaladze et al., 2009; Freund et al., 2006, 2009; Freund, 2011). Toutain and Baubron (1998) reported that the radon and other gases from the earth's crust near the active fault progress toward the atmosphere and cause ionization. The increased radon release produces a non-pronounced heat release
(increasing air temperature) in the atmosphere by connecting the water molecules to the ions. This increase in air temperature leads to variability in air conductivity (Pulinets et al., 2006). The amount of electron density in the ionosphere increases/decreases by this chaining process. Freund et al. (2006) detected the ionization of the side surfaces of the block where the air was ionization by increasing the





mechanical pressure applied to the upper surface of a granite block in the laboratory. With this
assumption, strains occurring in the huge rocks in the lithosphere before the earthquakes can cause
electron emission towards the atmosphere and may cause changes in the ionosphere (Freund et al., 2009).

In this study, the temporal, spatial, and spectral analysis was applied to the GNSS based TEC data to
detect ionospheric anomalies before the Mw 7.3 Iran-Iraq border earthquake on November 12, 2017. The
STFT and a running median process were applied to define abnormalities in the TEC time series. The
indices Kp, Dst, F10.7, IMF Bz, Ey, and $V_{SW}$ were also analyzed to show the effect of space weather
conditions on TEC variation. The paper is organized as follows: In Section 2.1, information on the Iran-
Iraq border earthquake is given. Section 2.2 includes data observations. In Section 2.3, GPS-TEC and
GIM-TEC data calculations are described. In Section 2.4, the methods used in the study are explained
capaciously. The results are given in Section 3, and Section 4 concludes the paper.

## 2   Data and Analysis

### 2.1 Iran–Iraq Border Earthquake

The deadliest earthquake of 2017, with at least 630 people killed and more than 8,100 injured occurred
near the Iran–Iraq border (34.911°N, 45.959°E) with a moment magnitude of 7.3 at a depth of 19.0 km on
November 12, 2017, at 18:18 UTC (U.S. Geological Survey, 2017). The earthquake was felt in Iraq, Iran,
and as far away as Israel, the Arabian Peninsula and Turkey. The focal mechanism of the earthquake is
pointed out as a thrust-faulting dipping at a shallow angle to the northeast. The earthquake occurred on the
continental collision between Eurasian and Arabian Plates located within the Zagros fold and thrust belt.

### 2.2 The GNSS based TEC data

The GNSS TEC data of five IGS stations and GIMs produced by the Center for Orbit Determination in
Europe (CODE) were used to investigate ionospheric anomalies before the Iran-Iraq border earthquake.
The location of the IGS stations and the epicenter are shown in Figure 1. The IGS stations are selected in
the earthquake preparation area, which is calculated by the Dobrovolsky equation, r = $10^{0.43M}$ km, where
M is the magnitude (Dobrovolsky et al., 1979). The earthquake preparation area of the Iran-Iraq border
earthquake is found to be 1380 km. The distance of IGS stations to the epicenter and other information are
given in Table 1. The geomagnetic coordinates of the stations were obtained from the KYOTO website
(http://wdc.kugi.kyoto-u.ac.jp/igrf/gggm/). Receiver Independent Exchange Format (RINEX) files of the
IGS stations were downloaded from the IGS website (ftp://igs.ensg.ign.fr/pub/igs/data/), and Ionosphere
Map Exchange Format (IONEX) files of CODE were downloaded from the National Aeronautics and
Space Administration (NASA) website (ftp://cddis.gsfc.nasa.gov/gps/products/ionex/). The CODE GIMs
covers $\pm 87.5^0$ latitude and $\pm 180^0$ longitude ranges with $2.5^0 \times 5^0$ spatial resolution (5184 cells) and 2-hour
temporal resolutions.





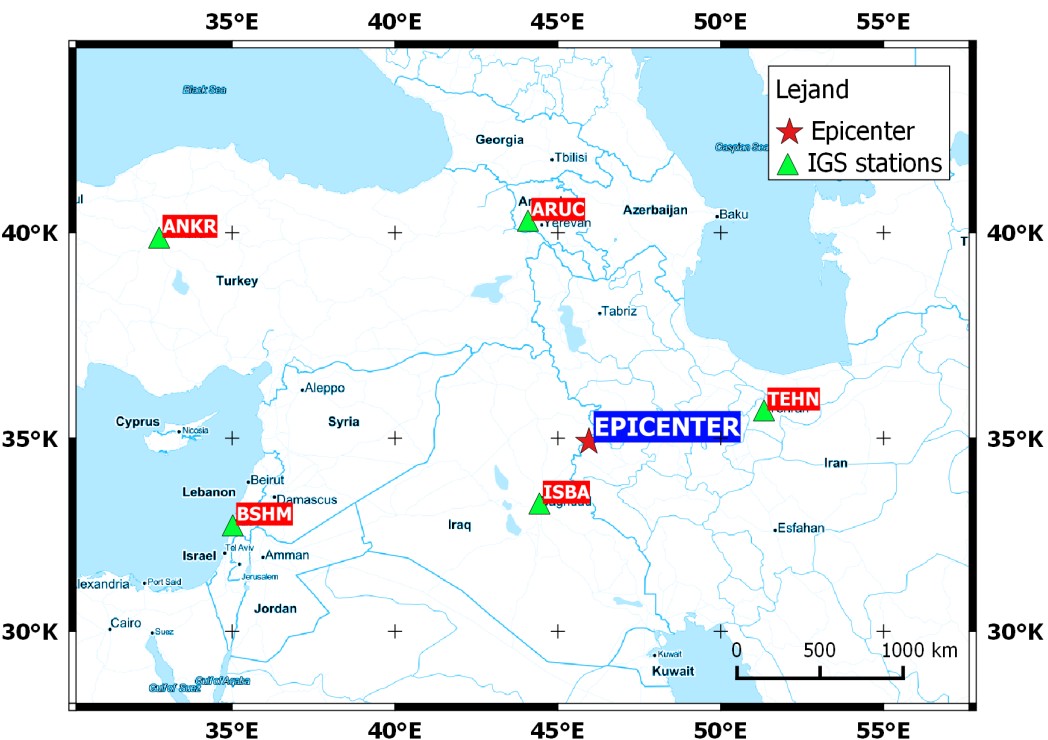

**Figure 1.** The epicenter of Mw 7.3 Iran-Iraq border earthquake and location of IGS stations (Map of the area is provided by http://maps.stamen.com and it was composed in QGIS program)

**Table 1** Information on the stations

| Site | Network | Country | Lat. ($^0$N) | Long. ($^0$E) | Geomag. Lat. ($^0$N) | Geomag. Long. ($^0$E) | Distance from the epicenter (km) |
|------|---------|---------|--------------|---------------|----------------------|-----------------------|----------------------------------|
| ANKR | IGS | Turkey | 39.8875 | 32.7583 | 36.54 | 112.72 | 1288.95 |
| ARUC | IGS | Armenia | 40.2856 | 44.0856 | 35.27 | 123.34 | 619.95 |
| BSHM | IGS | Israel | 32.7789 | 35.0200 | 29.23 | 113.25 | 1037.09 |
| ISBA | IGS | Iraq | 33.3414 | 44.4383 | 28.40 | 122.24 | 223.72 |
| TEHN | IGS | Iran | 35.6972 | 51.3339 | 29.79 | 129.11 | 495.45 |

The TEC describes the number of free electrons in a cylinder with 1 m$^2$ base area throughout the line of sight (LOS). The unit of the TEC (TECU) is equal to $10^{16}$ electron/m$^2$. The linear integral of the electron density along the signal path ($\int_l Ne(\vec{r}, t) ds$) corresponds to the Slant Total Electron Content (STEC). STEC depends on the signal path geometry from GNSS satellites (above 20.000 km height from the earth's surface) to a receiver. STEC is converted to the Vertical Total Electron Content (VTEC) with a


mapping function. This conversion provides the number of free electrons perpendicular to the earth. VTEC is used for the input data of the global and regional ionosphere models, and it is a more useful parameter to define all ionization in the ionosphere. Assuming all electrons are gathered in a thin layer, TEC values in the receiver's azimuth is obtained by the weighted average of the VTECs of all visible satellites (Schaer, 1999).

The effect of the ionosphere to the GNSS signal is directly proportional to the number of free electrons throughout LOS and inversely proportional to the square of the frequency of the GNSS signals (Hofmann-Wellenhof et al., 1992). The TEC parameter can be calculated with at least two different frequencies of GNSS signals because the effect of the ionosphere during the signal transition depends on the signal frequency. In recent years, the TEC parameter is obtained from single-frequency receivers by Precise Point Positioning (PPP) technique in which some parameters in the TEC calculation model are derived from IGS (Hein et al., 2016; Li et al., 2019). In this study, the Geometry-Free Linear Combination ($L_4 = L_1 - L_2$) and "leveling carrier to code" algorithm is used to calculate TEC values of five IGS stations (Ciraolo et al., 2007). $L_4$ combination of carrier phase and code observations are as follows,

$$L_4 = L_1 - L_2 = -\alpha \left( \frac{1}{f_1^2} - \frac{1}{f_2^2} \right) STEC + \lambda_1 B_{1,i}^k - \lambda_2 B_{2,i}^k \tag{1}$$

$$P_4 = P_1 - P_2 = \alpha \left( \frac{1}{f_1^2} - \frac{1}{f_2^2} \right) STEC + c(\Delta b^k - \Delta b_i) \tag{2}$$

where α is a constant, f is the signal frequency, $\lambda B_i^k = \lambda \left( N_i^k + \delta N_i^k \right) + c(b^k + b_i)$ is the initial phase ambiguity (i and k indexes refer to receiver and satellite respectively), λ is the wavelength, $N_i^k$ is an integer, $\delta N_i^k$ is the effect of the phase wind-up, c is the speed of light, $b^k$ is the satellite, and $b_i$ is the receiver hardware delays (DCB: Differential Code Biases). The phase leveling technique is based on differences carrier phase and code observations on a continuous arc to reduce ambiguities from the carrier phase ($L_4$).

$$\langle L_{4,arc} + P_4 \rangle_{arc} \cong \lambda_1 \delta N_1 - \lambda_2 \delta N_2 = B_4 \tag{3}$$

$$L_4 = L_4 + \langle L_{4,arc} + P_4 \rangle_{arc} = \alpha \left( \frac{1}{f_1^2} - \frac{1}{f_2^2} \right) STEC + b_4^k + b_{4,i} + B_4 \tag{4}$$

In Eq. 3, the carrier phase observations are leveled with a bias produced by phase ambiguity. Finally, the STEC is calculated using Eq. 5.

$$STEC = \alpha \left( \frac{1}{f_1^2} - \frac{1}{f_2^2} \right)^{-1} \left( L_4 - \left( B_4 + b_4^k + b_{4,i} \right) \right) \tag{5}$$

The STEC is converted to VTEC using the Single-Layer Model and a mapping function.

$$VTEC = STEC \sqrt{1 - \left( \frac{R_E}{R_E + h_m} \right)^2 \cos^2 \varepsilon} \tag{6}$$

To define the number of free electrons in the receiver's azimuth, TEC is generally calculated by the weighted average of the VTECs of all visible satellites (Çepni and Şentürk, 2016).



$$TEC = \frac{\sum_{i=1}^{N} W_i VTEC_i}{\sum_{i=1}^{N} W_i}\Bigg|_{T_1}^{T_2}; \text{ } T_1\text{-}T_2 \text{ is time-lapse interval} \tag{7}$$

where $W_i$ indicates the weight of a satellite, which is generally described as a component of the satellite elevation angle, $i = 0,1,\ldots,n$ and $n$ is equal to the number of visible satellites at any epoch.

TEC values of the epicenter are interpolated from the nearest four grid points of GIMs using a simple 4-point bivariate interpolation (Schaer et al., 1998).

$$TEC(\lambda_e, \beta_e) = |1 - m \quad m| \begin{vmatrix} VTEC_{00} & VTEC_{01} \\ VTEC_{10} & VTEC_{11} \end{vmatrix} \begin{vmatrix} 1 - n \\ n \end{vmatrix} \tag{8}$$

$$m = |\lambda_e - \lambda_0|/\Delta\lambda_{GIM} \tag{9}$$

$$n = |\beta_e - \beta_0|/\Delta\beta_{GIM} \tag{10}$$

where $m$, $n$ are latitudinal/longitudinal scale factor, $\beta_e$ and $\lambda_e$ is geocentric latitude/longitude of the epicenter, $\beta_0$ and $\lambda_0$ is geocentric latitude/longitude of the nearest grid point, $\Delta\beta_{GIM}$ and $\Delta\lambda_{GIM}$ are spatial resolutions of the latitude/longitude of the GIMs, $VTEC_{00}, VTEC_{01}, VTEC_{10}, VTEC_{11}$ are VTECs of the nearest grid points.

### 2.3 The Short-Time Fourier Transform and Running Median Methods

The STFT is a method of obtaining the signal frequency information in the time domain as a modified version of the classical Fourier (Gabor, 1946). The STFT provides the analysis of a small part of the signal at a particular time with the "windowing" technique (Burrus, 1995). The method divides the signal with a fixed time-frequency resolution (the size of the window is fixed in all frequencies) and presents the results in the time-frequency domain. It provides information about both when and at which frequencies a signal occurs. In this way, the method can provide statistical information about where and when the abnormality occurs in a TEC time series. The STFT of a signal is calculated by Eq.11.

$$STFT(\tau, f) = \int_{-\infty}^{+\infty} f(t)g(t - \tau)e^{-i\omega t} \, dt \tag{11}$$

where $f(t)$ is a time series (e.g., TEC), $g(t)$ is the window function, $\tau$ is a shifting time variable, and $\omega$ is the angular frequency. Here, a discrete STFT that provides identify and collect the frequency anomalies in the time domain was applied to obtain a time-frequency map of the TEC variation. The Gaussian window was also used as the window function $g(t)$ (Harris, 1978).

$$g(t) = e^{-0.5\left(\alpha \frac{t}{(N-1)/2}\right)^2} \tag{12}$$

where $N$ is the length of the window, and $\alpha$ could be termed as a frequency parameter. The width of the window is inversely related to the value of width factor ($\alpha$), and the $\alpha$ parameter, which controls the frequency resolution at both extremities, was taken as 0.005 in this study. When $\alpha$ value increases, the window becomes narrower, so the selected $\alpha$ parameter gives relatively accurate resolution in the frequency domain (see Fig.1).

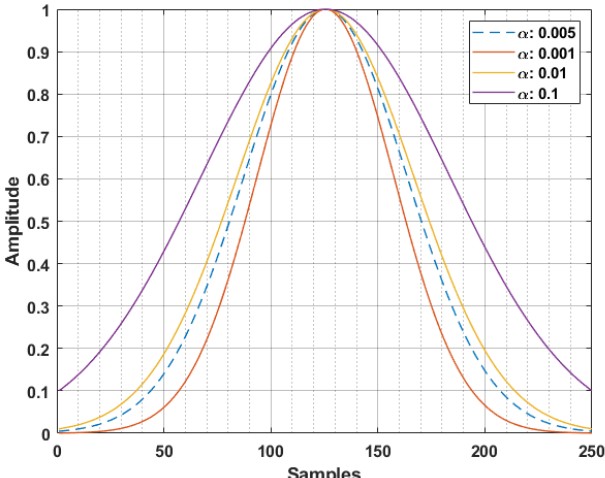

**Figure 2.** Gaussian windows functions according to α parameter

A well-known anomaly detection method (running median) for seismoionospheric studies was used to validate STFT results. This method is based on distribution moments median (M) and standard deviation (σ). In our analysis, the median of TEC values in the previous 15 days was calculated to find the divergence from the observed TEC on the 16th day. The lower (LB) and upper (UB) bounds were calculated by Eq.13-14 to assign the level of the divergence.

$LB = M - 2\sigma$          (13)

$UB = M + 2\sigma$          (14)

    When observed TEC of the 16th day is exceeded UB or LB, the positive or negative abnormal TEC signal is approved, respectively. The observed TEC between the UB and LB indicates no abnormal condition in the ionosphere. Assuming TECs are in a normal distribution with mean µ and standard
deviation σ, the divergence of 2σ declare that ionospheric phases are detected with a confidence level of about %95.

    The percentage of divergence degree of TEC (DTEC) was also calculated by the deviation from median values in GNSS TEC analysis. Since DTEC provides the relative TEC, it is more successful in detecting abnormalities at dusk when TEC values are lower.

$DTEC = [TEC_{observed} - TEC_{median}] \times 100/TEC_{median}$          (15)

### 3   Results

### 3.1 Space Weather Before the Earthquake

The space weather indices Kp, Dst, F10.7, IMF Bz, Ey, and $V_{SW}$ were cross-checked with TEC times series to reveal the effects of space weather on TEC disturbances. The indices obtained from the OMNI





website (https://omniweb.gsfc.nasa.gov/form/dx1.html). The time series of the indices with 15 days before the earthquake were given in Fig. 3.

In Fig. 3a, IMF Bz, and Ey indices have some fluctuations on 1-2 October and 7-11 October. These two indices remained calm on other days. In Fig. 3b, the $V_{SW}$ index increased rapidly from 300 km/s to 650 km/s on October 7. On the same day, the Dst index also decreased from +30 nT to -70 nT (see Fig. 205    3c). In both indices indicates a moderate magnetic storm (G2 level, Kp=6) occurred on the 7th of October. On the other days, it was determined that the indices values were at levels where atmospheric conditions to be considered calm. In Fig 3d, F10.7 and Kp indices were shown. F10.7 values continue to be quiet (<80 sfu) along 15 days before the earthquake. The index ranges from 65-75 sfu. Kp values indicate the disturbed magnetic condition between 7-11 October, whereas other days have no magnetic activity values. 210    Fig. 3 suggests that the moderate magnetic storm that occurred five days before the earthquake was capable until the one days before the earthquake. The fluctuations in IMF Bz and Ey indices on 1-2 October were not seen in other indices. The other days are quite calm in terms of space weather.

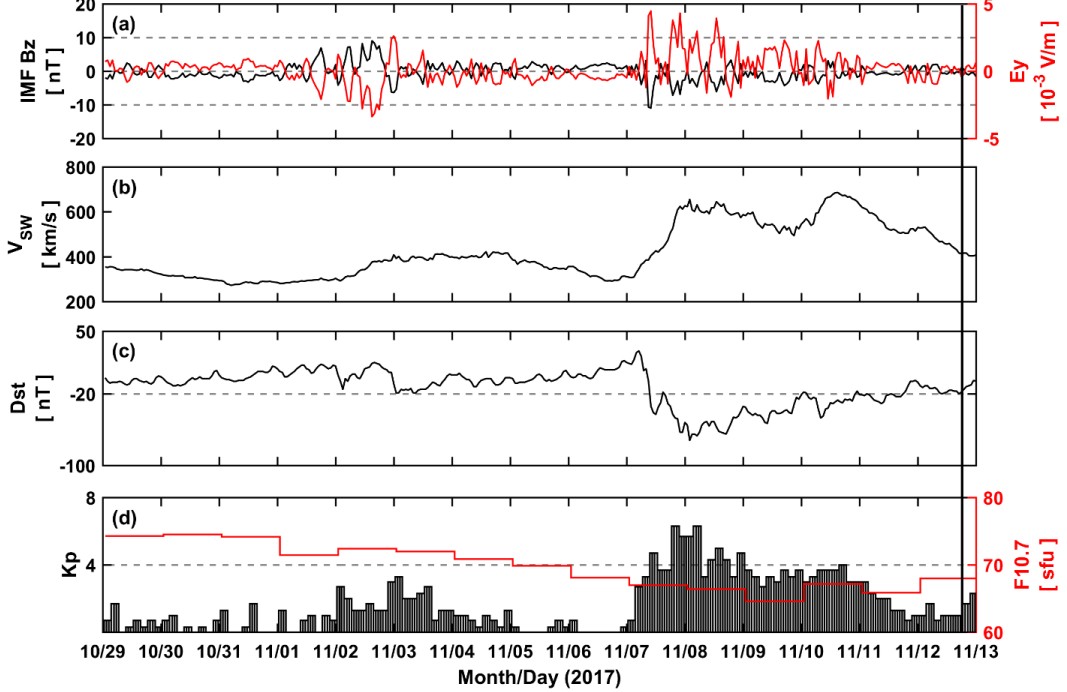

**Figure 3.** (a) IMF Bz and Ey (b) $W_{SW}$ (c) Dst (d) Kp and F10.7 indices before 15 days of the earthquake. 215    The vertical black line indicates the earthquake time.

### 3.2 Temporal and Spectral TEC Variation of GNSS Observations

TEC values over the epicenter location (34.911°N, 45.959°E) were obtained by interpolation from the vTEC values of the four grid points nearest to the epicenter in the GIMs to reveal ionospheric





abnormalities in the zenith of the epicenter. The anomalies were detected by the running median method based on median and ±2 standard deviations. In Fig. 4a observed and median TEC, upper/lower bounds were shown from 29 October to November 12, 2017. The anomalies were shown in Fig. 4b. There were 1-2 TECU positive anomalies on November 3-4 and some small positive/negative anomalies 1-6 days before the earthquake.

In Fig.5, the GNSS based TEC time series of ANKR, ARUC, BSHM, ISBA, and TEHN were demonstrated. The sampling rate of TEC data is 30 seconds. The stations were selected within the earthquake preparation area to reveal the earthquake-induced TEC fluctuations on TEC variation. The results showed that positive anomalies were detected on November 3-4, 2017, with 1-4 TECU in all stations. Some positive/negative anomalies were also determined on November 7-12. These anomalies should be related to the moderate magnetic storm that started on 7 November (the main phase of the storm occurred on 8 November).

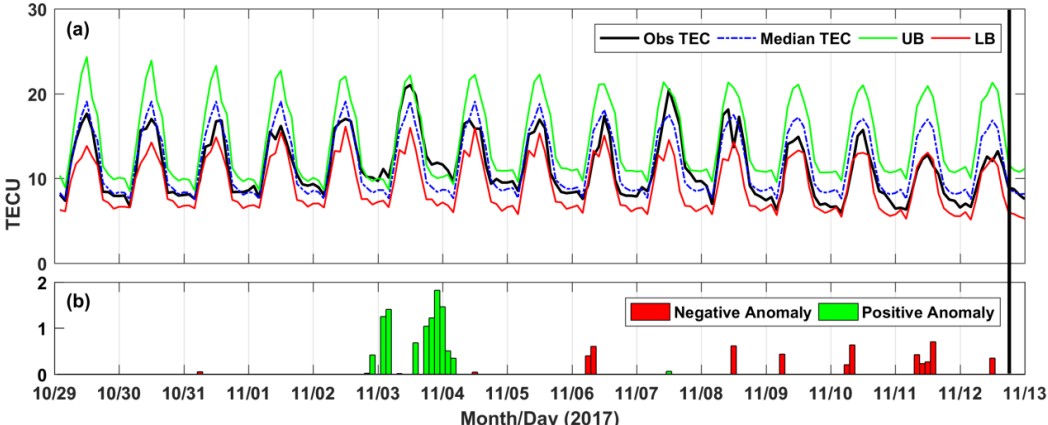

**Figure 4.** (a) TEC values of CODE GIMs over the epicenter (b) positive and negative anomalies. The vertical black line indicates the earthquake time.



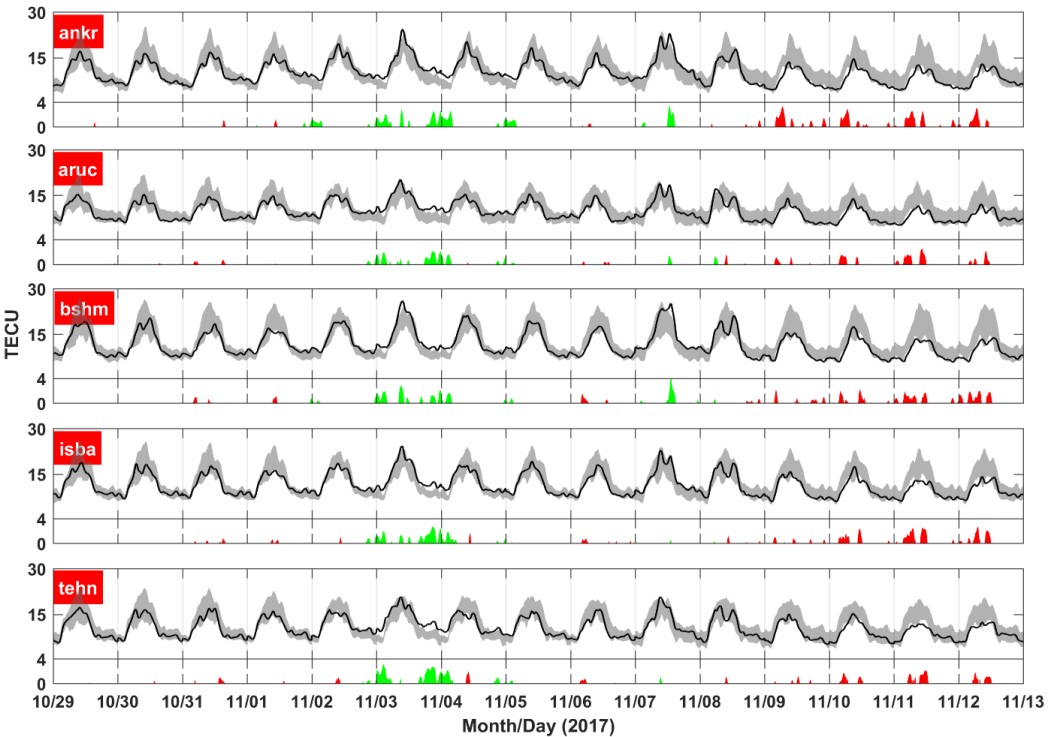

**Figure 5.** GNSS TEC variation of the five IGS stations where locate in the earthquake preparation area. The solid black lines indicate TEC values of stations, and the gray areas demonstrate M ± 2σ. The positive and negative anomalies were shown by green/red areas, respectively.

In Fig. 6, DTEC data of five IGS stations are given. DTEC reveals the relative change of observed TEC values to the median TEC value. The ionosphere has a significant day-to-day variability due to thermospheric dynamics even though quiet space weather. The diurnal TEC variation related to the lower atmosphere usually does not exceed ±30% according to the background TEC data (Forbes et al., 2000; Mendillo et al., 2002). In Fig. 6, we showed the 30% limit in the green area. Accordingly, DTEC values remaining in the green space can be accepted as the changes due to the daily day-to-day variability of the ionosphere. It was observed that the 30% limit was exceeded in the positive direction on November 2-5 and 7, in the negative direction between 8-12 November. The highest positive DTEC was detected on November 4 with + 62.5% and the lowest DTEC on November 9 with -43% at the ANKR station.

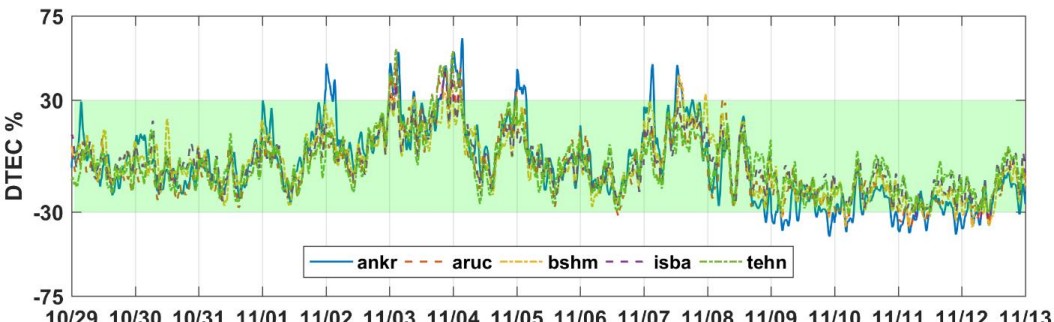

**Figure 6.** DTEC values of five IGS stations.

The STFT method was applied as a spectral analysis of GNSS based TEC data with a 30-second
sample rate. The method provides TEC anomalies both in the time and frequency domains. The amplitude
value ranges from 0 to 30 TECU. The STFT results are shown in Fig. 7. At the ANKR station, high
amplitude values are seen from November 2 to November 5 and November 7. The highest amplitude value
of about 30 TECU was seen on November 3. At the ARUC station, high amplitudes were seen all day on
November 3. This station has a relatively smaller amplitude (~24 TECU) value than the other stations. At
the BSHM station, high amplitudes are seen on November 3 and 7. In this station, the highest amplitude
value of 29.5 TECU was seen on November 7. At the ISBA and TEHN stations, the high amplitudes were
recognized on November 3. The highest amplitudes are between 27-30 TECU. In all stations, the largest
variations of the TEC anomalies correspond to smaller frequencies ($\leq 0.5 \times 10^{-5}$ Hz), and the maximum
amplitudes are between 25 and 30 TECU. The STFT analysis had a high amplitude on the days of
anomalies, which is defined in the running median process. Therefore, the results of STFT are well-
correlated with classical methods. The fact that the STFT method reveals TEC anomalies without any
background value is the strength of the method versus classical methods.

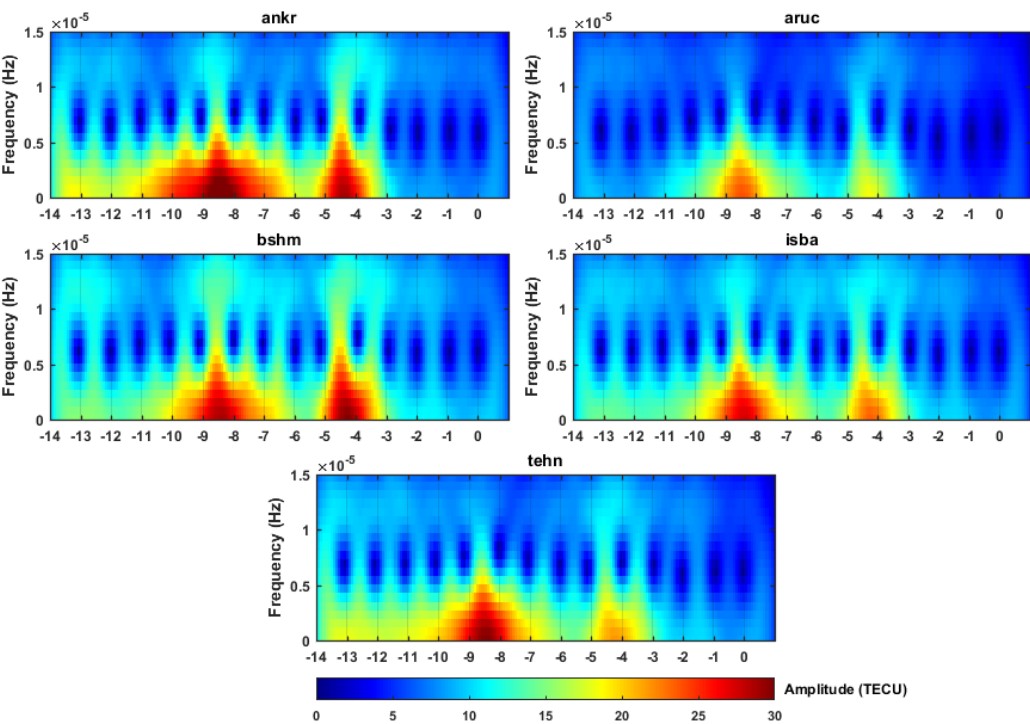

**Figure 7.** STFT analysis of GNSS TEC data of five IGS stations. The x-axis shows pre-earthquake days.

**3.3 Spatial Analysis of Abnormal Periods of TEC Variation**

The remarkable abnormal days (3, 4, 7, and 8 November) detected in the temporal and spectral analysis were spatially investigated by anomaly maps, which are created with CODE GIM data. These anomaly maps bounded by $60^0$ N-$60^0$ S latitudes, $180^0$ W-$180^0$ E longitudes, and have a temporal resolution of 2-hours. In maps, the epicenter of the earthquake is shown in a purple circle. The TEC anomalies in the
270 anomaly maps were detected by the running median method based on M $\pm$ 2σ. In Fig. 8, the anomalies range between -5 and +7 TECU on November 3-4. Fig. 8 showed that anomaly areas were locally distributed and a notable anomaly area concentrated near the earthquake epicenter. This area located toward the Northeast side of the epicenter with 2-5 TECU from 14:00 UTC to 02:00 UTC on November 3-4. An anomaly area also located on the Southeast side of the epicenter with 6-7 TECU between 04:00
and 06:00 UTC on November 4. These anomalies are interesting because no other anomaly region is seen in a large area, and it is located only in close areas to the epicenter. In Fig. 9, the anomalies range between -6 and +14 TECU on November 7-8. The only remarkable detail here is that the anomalies are distributed globally, as opposed to Fig. 8. The changes detected in the relevant days mostly point to an ionospheric variation caused by a magnetic storm.





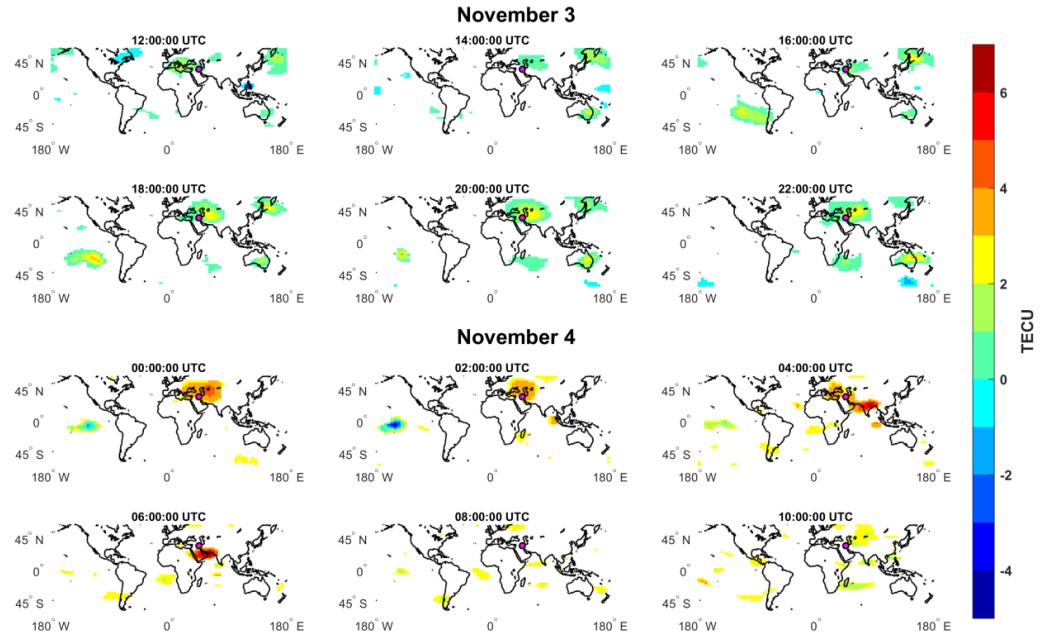

**Figure 8.** The anomaly maps on November 3-4, 2017.

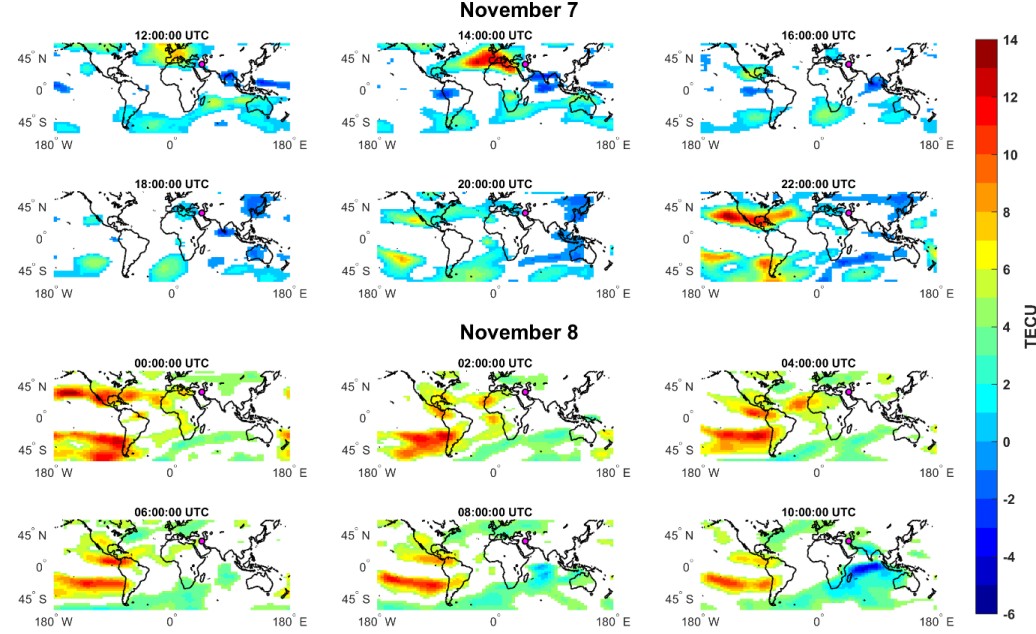

**Figure 9.** The anomaly maps on November 7-8, 2017.





It is reasonable to argue that anomalies that occur in the nighttime in the period of calm space weather may be related to the earthquake or other phonemes because the solar penetration towards the ionosphere reduces in the night. Therefore, the detected anomalies between 18:00 UTC (21:00 LT) and 02:00 UTC (05:00 LT) on November 3-4 should be the precursor of the Iran-Iraq border earthquake due to dusk time, quiet space weather and local distribution.

## 4    Conclusion

The TEC data of CODE GIM and five IGS stations were analyzed to reveal the earthquake-induced ionospheric anomalies of the Mw 7.3 Iran-Iraq border earthquake. For this purpose, a classical method named as running median and STFT method were applied to the TEC time series from October 29 to November 13, 15 days before the earthquake. The running median process of TEC variation was shown considerable positive anomalies as 1-4 TECU on November 3-4. This value is outlined from the mean of a normal distribution with a width of two standard deviations that is defined as a 95% confidence level. These positive anomalies were also detected in the spectral analysis. The STFT method was used for spectral analysis. STFT is a powerful tool for processing a time series without any background values (mean, median, quiet days, etc.). Independence from background data minimizes the error sources of these data (other unexpected changes, main trends of the ionosphere such as annual, semi-annual, and seasonal). The results showed the power of the STFT method in the detection of TEC anomalies.

There are some positive/negative anomalies 1-6 days before the earthquake, but these anomalies should be caused by a moderate geomagnetic storm on November 7-8. A geomagnetic storm affects the ionosphere as a whole, producing more global variations of TEC compared to the localized phenomena of seismoionospheric coupling. In Fig. 9, the global TEC changes of the moderate magnetic storm is seen. On the contrary, the anomalies occurring on 3-4 November, which are thought to be caused by the earthquake, have local distribution, and are concentrated near the epicenter (see Fig. 8).

Fig. 10 showed the prompt penetration electric fields (PPEFs) at $46^0$ E longitude (geographical longitude of the epicenter) on 3-4 November and 7-8 November. The PPEFs are observable in the ionosphere immediately after being transported to the magnetosphere by the solar wind (Tsurutani et al., 2008). The PPEFs also occur during the negative values of IMF Bz (Astafyeva et al., 2016). Fig. 3 indicated an increase of the solar wind from 300 km/s to 650 km/s, and the IMF Bz decreased to negative values as about -10 nT. Accordingly, fluctuations in PPEF variation are observed between 06:00 UTC and 02:00 UTC on November 7-8 (see Fig. 10b). Many studies have reported that PPEFs cause positive and negative phases in the ionosphere during magnetic storms (Basu et al., 2007; Tsurutani et al., 2008; Mannucci et al., 2009; Lu et al., 2012; Astafyeva et al., 2016). Fig 10b indicated that the moderate magnetic storm caused the positive and negative anomalies in the ionosphere along with the change in PPEF values on 7-8 November. On the contrary, no significant difference in PPEF values was observed in Fig. 10a. These PPEFs values indicated that a magnetic storm or solar wind could not affect the TEC variation on 3-4 November.

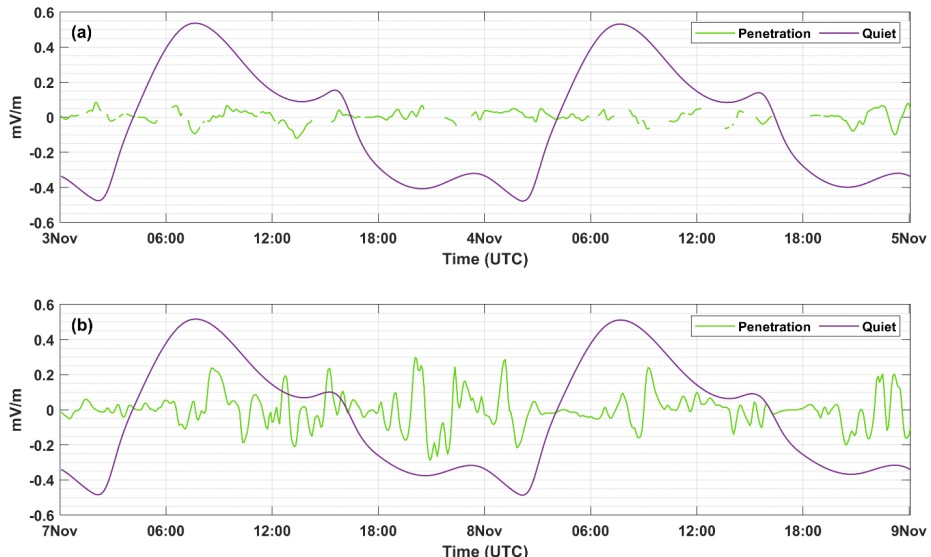

**Figure 10.** The prompt penetration electric fields at $46^0$ E longitude (a) on November 3-4 (b) on November 7-8, 2017.

Although the space weather is rather quiet on 3-4 November, the DTEC values of the five IGS stations exceeded the 30% limit corresponding to the day-to-day variability of the ionospheric TEC and reached

65%. This value indicates remarkable positive ionospheric anomalies. It can be said that the positive anomalies 8-9 days before the earthquake should be associated with the Iraq-Iran border earthquake because they occurred in the close areas to the epicenter and dispersed in local rather than global. Also, the anomalies continued all day, detecting at all IGS stations which are near the epicenter.

This study showed the advantages of using different approaches to detect earthquake-related
anomalies. Notably, it will be useful to prefer spectral analysis methods for the anomaly detection process as a new and promising approach in future studies.

*Data availability.* The RINEX files of the IGS stations are publicly available at the IGS website ftp://igs.ensg.ign.fr/pub/igs/data/, the IONEX files of CODE are publicly available at the NASA website
ftp://cddis.gsfc.nasa.gov/gps/products/ionex/, and the space weather indices are publicly available at the OMNI website https://omniweb.gsfc.nasa.gov/form/dx1.html.

*Author contributions.* ES carried out the data analysis, prepared the plots, and interpreted the results. SI provided processed GIM based TEC time series. IS interpreted the storm-time effects on the ionosphere. ES prepared the manuscript with contributions from all authors.

*Competing interests.* The authors declare that they have no conflicts of interest.



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
