# Peer review of "Ionospheric Anomalies Associated with Mw7.3 Iran-Iraq Border Earthquake and a Moderate Magnetic Storm"

_Annales Geophysicae, 2020_

## Referee Comment (RC1) · Anonymous Referee #1 · 1 Jun 2020

**The review of "Ionospheric Anomalies Associated with Mw7.3 Iran-Iraq Border Earthquake and a Moderate Magnetic Storm" by E. Şentürk et al.**

Data processing techniques are utilized to investigate total electron content (TEC) abnormal variability prior to the Mw7.3 Iran-Iraq Border earthquake. TEC data were analyzed along with geomagnetic condition coefficient fluctuations. Based on the results, authors propose that anomalies 8-9 days prior to the earthquake relate to the precursory activity.

Generally, ionospheric precursory methodologies still remain a debatable topic, nevertheless increasing interest from scientific society and rocketing of publications in recent time. In my opinion, the main problem with the used here methodology lays in a lack of the analysis of anomaly appearances on long-time intervals (Major comment 4,5). It is hard to assess the validity of the methodology having observations only 2 weeks prior to the earthquake. Also, authors do not provide error analysis, what makes it difficult to understand how anomalies of 1-4 TECu are far from TEC accuracy threshold (Major comment 1-3). Thus, although the results deserve the publication, I would suggest some major revisions that in my opinion are required for better understanding of the material and clarification of subtle moments.

**SPECIFIC COMMENTS (MAJOR COMMENTS)**

1. As of my awareness, CODE provides interpolated (spherical harmonic fitted) TEC maps (please provide a citation at L105). This may result in biases generated by data interpolation. What accuracy is expected for the derived vTEC values over the epicenter based on CODE TEC maps? Why authors found it is necessary to analyze interpolated CODE maps, instead of just considering 5 available stations (Table 1) and calculating TEC over epicentral position with them? Also, does CODE use the same IGS stations in the considered region to produce vTEC maps? If so, authors analyze the same data twice (e.g., Figure 4 and 5). Please, clarify which stations in the considered region are used by CODE. Again, how good "anomaly maps" are for the estimation of absolute deviations (as they also based on CODE GIM interpolated data)? Were they cross-checked with vTEC over the epicenter calculated based on 5 stations? Do values agree?

2. Authors introduce satellite and receiver biases (eq. 2), but do not indicate if these biases were corrected. It is not clear what methodology is used for the correction of these biases and what errors are expected for the determination of vTEC. The analysis and incorporation of these biases is an important factor while discussing the variations of absolute vTEC and I believe this should be clarified in the text.

3. Authors provide the equation for the calculation of TEC averaged from all satellites (eq. 7). However, it is not clear if all Ionospheric Pierce Points (IPP)

for used observations were over the earthquake preparation area (determined as 1380 km). If authors carry out the selection of TEC observations outside this area, it seems possible that found anomalies results from the area outside of it. For eq. 6, it is not clear what ionospheric shell height is used for the calculation of vTEC. Also, it is not clear what elevation angle cut-off is used for vTEC observations based on eq. 7.

4.  In my opinion, authors use very narrow range of days and only prior to the earthquake (from 10/29 to 11/13). It is crucial to understand whether positive anomalies appear only before the earthquake or on a constant basis during quite times. Such analysis requires additional processing of data before and after the earthquake. I would consider range between -3/+3 months, along with the analysis of geomagnetic indexes and the use of the same stations over the same region.

5.  Authors reference publications by Forbes et al., 2020 and Mendillo et al. 2002, but they do not explicitly mention that TEC observation variability cannot exceed 30%, incorporating possible satellite and instrumentation biases as well as integrated nature of TEC, IPP locations, recalculation of vTEC from sTEC etc. Also, Forbes et al., 2000 discuss "high frequency" variability of 25-35% under quite Kp index < 1, whereas (according to Figure 3), Kp index on $3^{rd}$ and $4^{th}$ of November seem to be higher than 1 (especially on $3^{rd}$ of November, where Kp index approaches 4). Do authors expect the same ~25-35% variability for Kp index of 4?

**TECHNICAL COMMENTS (MINOR COMMENTS)**

1.  Please, clarify the choice of the window for Gaussian function as 0.005. What period it corresponds?
2.  Consider using the same x-axis on all plots (e.g., on Figure 7, there are days prior to the earthquake, although on Figure 5 there are Month/Day). Also, authors may want to indicate periods instead of frequencies, as it is difficult to assess the period from ~10^-5 Hz).
3.  From Figure 5, I didn't find anomalies up to 4 TECu, nor from Figure 4 (as stated in the Conclusion). Please, clarify what is a maximum absolute deviation/anomaly value found and if it is higher than expected threshold for the calculation of vTEC.
4.  Authors may consider moving Figure 10 and appropriate discussion to Section 3, instead of discussing data analysis results in the Conclusion.
5.  Please, consider introducing all abbreviations in the text (not only in the abstract), e.g., LMTF, CMONOC, IGS, GIM etc., along with indexes in paragraph 80 (IMF, Ey, Vsw).
6.  Please revise paragraphs 25-60, as they discuss studies that are related to both post-seismic (acoustic-gravity driven disturbances in the ionosphere) and pre-seismic activity. These are 2 completely different fields of studies

and this should be clarified for readers not familiar with the topic (instead, the discussion of post-seismic studies may be fully excluded from the text).

7. Paragraph 205-210 – Should it be November instead of October?
8. Figure 1 – Should it be the indication of the northern hemisphere latitudes as N (not K)?
9. Why abnormal TEC variations are seen 8-9 days after the earthquake and not in closer dates? What is a physical explanation authors may suggest for this?

---

## Author Comment (AC1) · 6 Jun 2020

**SPECIFIC COMMENTS (MAJOR COMMENTS)**

1. As of my awareness, CODE provides interpolated (spherical harmonic fitted) TEC maps (please provide a citation at L105). This may result in biases generated by data interpolation. What accuracy is expected for the derived vTEC values over the epicenter based on CODE TEC maps? Why authors found it is necessary to analyze interpolated CODE maps, instead of just considering 5 available stations (Table 1) and calculating TEC over epicentral position with them? Also, does CODE use the same IGS stations in the considered region to produce vTEC maps? If so, authors analyze the same data twice (e.g., Figure 4 and 5). Please, clarify which stations in the considered region are used by CODE. Again, how good "anomaly maps" are for the estimation of absolute deviations (as they also based on CODE GIM interpolated data)? Were they cross-checked with vTEC over the epicenter calculated based on 5 stations? Do values agree?

- The main purpose of using GIM TECs was to validate the calculated GNSS TECs with a reliable data. Now, we have expanded this validation even further (the analyzes and results in below are not included in the article. It was only carried out in response to your question.)
- We estimated the CODE GIM vTEC values at the location of the GNSS stations (ankr, aruc, bshm, isba and tehn) similar to the vTECs of the epicenter. The RMSE and Bias values are seen in the Fig.1. RMSE values range between 0.52-0.68 TECU and Bias values range between 0.01-0.44 TECU.
- The RMSE and Bias values indicated that GIM vTEC values of CODE compatible with GNSS TECs of IGS stations. These values also prove the accuracy of "anomaly maps".
- A citation added for CODE GIM at L100.

[Figure]

**Figure 1** GNSS TECs and GIM vTECs at the location of IGS stations (left panel), differences between GNSS and GIM, RMSE and Bias values (right panel).

2. Authors introduce satellite and receiver biases (eq. 2), but do not indicate if these biases were corrected. It is not clear what methodology is used for the correction of these biases and what errors are expected for the determination of vTEC. The analysis and incorporation of these biases is an important factor while discussing the variations of absolute vTEC and I believe this should be clarified in the text.

- We used the TEC time series of IGS station so DCBs were obtained from daily IONEX files.
- We added a new sentence in L130. *"The DCBs of satellites and receivers are available in the daily IONEX files for IGS stations, but receiver DCBs of no-IGS stations must be calculated in the TEC calculation process."*

3. Authors provide the equation for the calculation of TEC averaged from all satellites (eq. 7). However, it is not clear if all Ionospheric Pierce Points (IPP) for used observations were over the earthquake preparation area (determined as 1380 km). If authors carry out the selection of TEC observations outside this area, it seems possible that found anomalies results from the area outside of it. For eq. 6, it is not clear what ionospheric shell height is used for the calculation of vTEC. Also, it is not clear what elevation angle cut-off is used for vTEC observations based on eq. 7.

- In the vTEC weighting process of obtaining TEC values of a station zenith, the VTEC value of a satellite with a low elevation angle already has low weight. TEC values are generally calculated as the VTEC value of a satellite with an elevation angle between $70^0$-$90^0$ (these IPPs are already very close to the station). Also, the earthquake preparation area is an empirical value and it is not absolute.
- We added a new sentence in L220. *"In the TEC calculation process, the satellite and receiver DCBs were obtained from IONEX files of CODE. The height of the single-layer was selected as 450 km and the elevation cut-off angle of 30° is taken."*

4. In my opinion, authors use very narrow range of days and only prior to the earthquake (from 10/29 to 11/13). It is crucial to understand whether positive anomalies appear only before the earthquake or on a constant basis during quite times. Such analysis requires additional processing of data before and after the earthquake. I would consider range between -3/+3 months, along with the analysis of geomagnetic indexes and the use of the same stations over the same region.

- We revised the Fig.4 as below. In the new version of Fig. 4, the 30 pre- and post-earthquake days were analyzed with the same method in the previous version.
- We also revised texts in all manuscript related to the change in Fig. 4.
- In Fig.4 (Fig.2 in this file), it was seen that anomalies occurred only on 3-4 November for quiet Dst (Dst > -20) during 60 days. Also, GIM and GNSS TEC values are in good agreement as stated in Q1, so we did not perform the same analysis for GNSS data. GNSS time series includes 15 days before the earthquake as the previous version of the article.
- We added a new sentence in L310. *"Only the CODE GIM time series were analyzed for 60 days, including 30 days before and after the earthquake. Thus, it has been revealed that the anomalies obtained are not a coincidence. Abnormalities are observed only on 3-4 November, when the Dst values represent quiet geomagnetic conditions (Dst > -20 nT)."*

[Figure]

**Figure 2** TEC values of CODE GIMs over the epicenter, positive/negative anomalies and Dst values during 30 pre- and post-earthquakes days.

5. Authors reference publications by Forbes et al., 2020 and Mendillo et al. 2002, but they do not explicitly mention that TEC observation variability cannot exceed 30%, incorporating possible satellite and instrumentation biases as well as integrated nature of TEC, IPP locations, recalculation of vTEC from sTEC etc. Also, Forbes et al., 2000 discuss "high frequency" variability of 25-35% under quite Kp index < 1, whereas (according to Figure 3), Kp index on 3rd and 4th of November seem to be higher than 1 (especially on 3rd of November, where Kp index approaches 4). Do authors expect the same ~25-35% variability for Kp index of 4?

- Please examine the anomalies in Fig. 5 and the situations exceeding ±30% in Fig. 6. You will notice the similarity. In Fig. 5, no-abnormal TEC conditions are determined with a 95% confidence level. DTEC values were within the 30% limit when GNSS TECs do not exceed upper or lower bounds. Here, the ±30% limit for DTECs represented quiet ionospheric conditions. However, I need to examine more data to reach a definitive conclusion about this. That's why I have cited some previous studies.
- In fact, it is important here not what Kp index values are, but whether TEC values represent abnormal conditions in the time series. The ±30% limit for DTEC is equivalent to the TEC value between the upper and lower limit (no-abnormal condition) in the running median method.
- We added a new sentence in L245. *"Fig. 6 also indicated that the ± 30% limits of DTEC variation are consistent with the no-abnormal condition of running median method (see Fig. 5)."*

**TECHNICAL COMMENTS (MINOR COMMENTS)**

1. Please, clarify the choice of the window for Gaussian function as 0.005. What period it corresponds?
- A value of 0.005 is a coefficient that controls the frequency resolution of the gauss window function. It is not related to the period. We used it because we provide the best resolution with this value.
- We added a new explanation in L170. *"Since it provided the best resolution, the α was chosen as 0.005 for this study."*

2. Consider using the same x-axis on all plots (e.g., on Figure 7, there are days prior to the earthquake, although on Figure 5 there are Month/Day). Also, authors may want to indicate periods instead of frequencies, as it is difficult to assess the period from ~10^-5 Hz).
- We revised the x-axis of Fig.7 as "Month/Day". The revised version of Fig. 7 is below.
- The y-axis in graphs indicate Fourier frequencies according to sampling rates of the TECs. So it is not necessary to convert them into periods.

[Figure]

3. From Figure 5, I didn't find anomalies up to 4 TECu, nor from Figure 4 (as stated in the Conclusion). Please, clarify what is a maximum absolute deviation/anomaly value found and if it is higher than expected threshold for the calculation of vTEC.
- We revised as 1-3 TECu in all text.
- These values statistically represent the values outside the 95% confidence intervals of the normal distribution curve. Even if we assume that there are systematic errors in GPS TEC values, these errors remain within the distribution curve.

4. Authors may consider moving Figure 10 and appropriate discussion to Section 3, instead of discussing data analysis results in the Conclusion.

- We opened a new section as Section 3.4 with a headline **"The Prompt Penetration Electric Fields (PPEFs) Variation in Abnormal Days"**. We added Figure 10 and related explanations to this section.

5. Please, consider introducing all abbreviations in the text (not only in the abstract), e.g., LMTF, CMONOC, IGS, GIM etc., along with indexes in paragraph 80 (IMF, Ey, Vsw).
- All abbreviations in the text were introduced.

6. Please revise paragraphs 25-60, as they discuss studies that are related to both post-seismic (acoustic-gravity driven disturbances in the ionosphere) and pre-seismic activity. These are 2 completely different fields of studies and this should be clarified for readers not familiar with the topic (instead, the discussion of post-seismic studies may be fully excluded from the text).
- We removed the paragraph from L24 to L29 which is including a lot of citation related to post-seismic (acoustic-gravity driven disturbances in the ionosphere) activity.

7. Paragraph 205-210 – Should it be November instead of October?
- The typo was corrected as "November".

8. Figure 1 – Should it be the indication of the northern hemisphere latitudes as N (not K)?
- The typo was corrected. The new version of Fig. 1 is below.

[Figure]

9. Why abnormal TEC variations are seen 8-9 days after the earthquake and not in closer dates? What is a physical explanation authors may suggest for this?
- We added possible physical explanations about the earthquake-ionosphere coupling to the introduction section of the article. There is not enough data in the study to make another comment on the physical explanation. Also, some researchers indicated the pre-earthquake ionospheric anomalies about 1-10 days before earthquakes (Xia et al., 2011; Inyurt et al., 2019, etc.).

---

## Referee Comment (RC2) · Anonymous Referee #2 · 15 Jun 2020

The manuscript presents an observational study of the ionospheric TEC precursors of the 12 November 2017 Iran-Iraq Border Earthquake. The study analyzed the TEC data from IGS stations surrounding the epicenter and the CODE GIMs using Short-time Fourier Transform method and a running median process. The study also analyzed space weather data to determine the contribution of geomagnetic activities to the TEC anomalies before the earthquake. The outcome of the study showed two groups of TEC anomalies with different causes: the anomalies 1-6 days before the earthquake were caused by a geomagnetic storm, while the anomalies 8-9 before the earthquake were related to the earthquake.

I find the manuscript fairly well-written in general. The study delivers interesting science results and would be inspiring to the community. In particular, the study presents a very nice demonstration of separating the space weather contribution from the earthquake contribution to TEC anomalies. However, there are certain ambiguities in methodology and results that need to be addressed, which are listed below.

1. The relation between the TEC anomalies on November 3-4 and the earthquake is weak given the evidence shown in the manuscript. The authors claim that the TEC anomalies on November 3-4 are earthquake precursors because of quiet space weather, local dispersion and proximity to the epicenter. Instead of quiet space weather, Figure 3 shows a mild geomagnetic activity on November 3-4, with elevated Kp comparing to days immediately before and after. Is it possible that the TEC anomalies on November 3-4 are due to this mild geomagnetic activity? To exclude this possibility, the authors have shown a) the localized anomaly on GIMs of November 3-4, and b) the negligible variations of prompt penetration electric fields on November 3-4.

For a), GIMs are interpolated GNSS TEC maps. It is not clear how many and where the GNSS stations are in generating the GIMs. Are the five IGS stations surrounding the epicenter included for the GIMs? To directly demonstrate that the TEC anomalies on November 3-4 are localized, why not show the lack of anomalies for IGS stations further away from the epicenter (outside of the earthquake preparation area), using the exact same methodology for analyzing the existing 5 stations? A few more panels on Figure 5 for other stations would say it all.

For b), I could not find how the PPEFs are calculated and what is the "Quiet" curve in Figure 10. Does the variation of PPEFs correlate with the TEC variations due to space weather? More explanation would be helpful.

2. Have the authors look into the wave characteristics, for instance the wave period/frequency and duration of the TEC anomalies on November 3-4? Are they similar to the characteristics of earthquake TEC precursors found in previous studies? This

would better support the argument that the TEC anomalies on November 3-4 are the earthquake precursors.

3. Line 15: molecules are separated into positively charged particles and electrons?

4. Second paragraph of Introduction: some of the references are for ionospheric anomalies during and after earthquakes, which has very different physical mechanisms from the earthquake precursors. I noticed that referee #1 has also pointed this out. I hope the authors successfully address this in the paper revision.

5. Line 46 and Line 79: GIM and STFT are not defined in the main text.

6. Line 95: Any references for CODE GIM?

---

## Author Comment (AC2) · 17 Jun 2020

The manuscript presents an observational study of the ionospheric TEC precursors of the 12 November 2017 Iran-Iraq Border Earthquake. The study analyzed the TEC data from IGS stations surrounding the epicenter and the CODE GIMs using Short-time Fourier Transform method and a running median process. The study also analyzed space weather data to determine the contribution of geomagnetic activities to the TEC anomalies before the earthquake. The outcome of the study showed two groups of TEC anomalies with different causes: the anomalies 1-6 days before the earthquake were caused by a geomagnetic storm, while the anomalies 8-9 before the earthquake were related to the earthquake.

I find the manuscript fairly well-written in general. The study delivers interesting science results and would be inspiring to the community. In particular, the study presents a very nice demonstration of separating the space weather contribution from the earthquake contribution to TEC anomalies. However, there are certain ambiguities in methodology and results that need to be addressed, which are listed below.

- Thank you for your favorable comments, your time and consideration.

1. The relation between the TEC anomalies on November 3-4 and the earthquake is weak given the evidence shown in the manuscript. The authors claim that the TEC anomalies on November 3-4 are earthquake precursors because of quiet space weather, local dispersion and proximity to the epicenter. Instead of quiet space weather, Figure 3 shows a mild geomagnetic activity on November 3-4, with elevated Kp comparing to days immediately before and after. Is it possible that the TEC anomalies on November 3-4 are due to this mild geomagnetic activity? To exclude this possibility, the authors have shown a) the localized anomaly on GIMs of November 3-4, and b) the negligible variations of prompt penetration electric fields on November 3-4.

For a), GIMs are interpolated GNSS TEC maps. It is not clear how many and where the GNSS stations are in generating the GIMs. Are the five IGS stations surrounding the epicenter included for the GIMs? To directly demonstrate that the TEC anomalies on November 3-4 are localized, why not show the lack of anomalies for IGS stations further away from the epicenter (outside of the earthquake preparation area), using the exact same methodology for analyzing the existing 5 stations? A few more panels on Figure 5 for other stations would say it all.

- We analyzed the TEC data of two stations outside the earthquake preparation area and presented the results in Figure 5. In addition, we revised Figure 1 and some sections in the article in accordance with the new situation. The revised version of Figure 1 and Figure 5 are in below.
- In addition, we explained the relationship between GNSS TEC and GIM TEC in the first part of the reply to Reviewer#1. Please check it.

[Figure]

Figure: Revised version of Figure 1.

[Figure]

Figure: Revised version of Figure 5.

For b), I could not find how the PPEFs are calculated and what is the "Quiet" curve in Figure 10. Does the variation of PPEFs correlate with the TEC variations due to space weather? More explanation would be helpful.

- We included the explanations about PPEFs as a separate section (Section 3.4 with a headline "The Prompt Penetration Electric Fields (PPEFs) Variation in Abnormal Days") with the recommendation of Reviewer#1, and made the first paragraph of this section according to your recommendation as follows.

- *"The PPEFs is the prompt reaction of the equatorial zonal electric field to solar wind alteration, which is component of the interplanetary electric field (IEF) and the equatorial zonal electric field (Manoj et al., 2008). The penetration part of PPEFs (green line in Fig. 10) is calculated by the interplanetary data which is provided by the OMNI web site. Also, the quiet (climatological) part of PPEFs (violet line in Fig. 10) is equal to the 81-day moving average of F10.7 cm solar flux (Manoj and Maus, 2012)."*

2. Have the authors look into the wave characteristics, for instance the wave period/frequency and duration of the TEC anomalies on November 3-4? Are they similar to the characteristics of earthquake TEC precursors found in previous studies? This would better support the argument that the TEC anomalies on November 3-4 are the earthquake precursors.

- I applied the STFT method to the TEC time series for the first time in the article related to Van EQ and achieved successful results similar to the results of Iran-Iraq EQ. We compared the success of the STFT with the classical method (running median). The results are consistent. The STFT only shows anomalies in the TEC time series. As known, more analysis is needed as was done in the study to establish the relationship between the anomalies and the earthquake.

*"Şentürk, E., Livaoğlu, H., Çepni, M. S. (2019). A Comprehensive Analysis of Ionospheric Anomalies before the M w 7· 1 Van Earthquake on 23 October 2011. The Journal of Navigation, 72(3), 702-720."*

3. Line 15: molecules are separated into positively charged particles and electrons?

- To make it more understandable, we revised this section as follows: *"When molecules are exposed to light energy emitted from the sun, their components are divided into atoms, which are electrons and a compact nucleus of protons and neutrons. Negatively charged electrons effect the propagation of electromagnetic signals traveling between space and earth."*

4. Second paragraph of Introduction: some of the references are for ionospheric anomalies during and after earthquakes, which has very different physical mechanisms from the earthquake precursors. I noticed that referee #1 has also pointed this out. I hope the authors successfully address this in the paper revision.

- We removed the paragraph from L24 to L29 which is including a lot of citation related to co-seismic and post-seismic (acoustic-gravity driven disturbances in the ionosphere) activity.

5. Line 46 and Line 79: GIM and STFT are not defined in the main text.

- We defined them in the new version of the manuscript.

6. Line 95: Any references for CODE GIM?

- A citation added for CODE GIM at L100.

---

## Referee Report (RR1)

**The revision of the manuscript "Ionospheric Anomalies Associated with Mw7.3 Iran-Irak Earthquake and a Moderate Magnetic Storm"**

After the first revision, the manuscript was substantially improved, clarifying data processing methodology and providing additional insight on the validity of outcomes made. I believe that the manuscript has already a potential to be published, though I think some minor clarifications should be made, along with technical corrections.

**Minor suggestions:**

1. Please, indicate which GNSS stations are used in CODE GIM maps. This question is related to the previous revision (Major comment 1). In Figure 1 of the "answers to reviewer" you showed comparison of CODE GIM and IGS, but does CODE use the same stations as chosen from IGS or different ones? If these are the same stations, what is a reason to provide the analysis based on CODE GIM interpolated maps if RINEX data for the same stations are available and discussed in the article? Also, for Figure 1 in "answers to reviewer" I cannot understand how the BIASes were calculated. There are definitely some peaks close or even reaching 2 TECu, which is comparable with the amplitude of the detected anomaly. For example, the negative peak for difference plot for station ANKR reaches ~2 TECu at 11/04. Please, clarify these points in the final text.

2. Please, indicate what accuracy of vTEC (absolute value) you expect in your calculations and how it was estimated. Also, as I wrote it earlier, Forbes et al., 2020 and Mendillo et al. 2002 do not discuss that TEC cannot exceed 30%, as it is now stated at L245. You may want to add this clarification to the text. For Figure 6, I would also suggest showing that 30% is consistent with no-abnormal conditions for the whole time period as shown in Figure 4. You may consider merging Figure 6 with Figure 3 or 4.

3. Why in Figure 4 I do not find the same strong negative anomalies 11/09-11/13 as in Figure 5? Also, some positive anomalies are shown for stations BSHM and ANKR at 11/07, but I can't find them in Figure 4. Generally saying, is there consistency between station analyses and CODE GIM maps? If not, what is a reason for inconsistencies and which data are better (this is some part related to equation 1 above)?

**Technical suggestions:**

4. Please, consider another word in the first sentence of the abstract rather than "popular".

5. 1st sentence of first paragraph – it is mentioned that the ionosphere is a dispersive layer. Dispersive for what? If you mean electromagnetic signals – please indicate, otherwise the sentence sounds incomplete.

6. Second sentence – what about ions? Please, consider rewriting this sentence.

7. 5th line – "to the Earth".

8. 5th line – I would write "To the first order, the degree of effect…."
9. 6th line – "free electrons"?
10. 8th line – please provide some references to daily, 27-day etc variations, I think that may provide reader better background.
11. Near 40 – Please clarify what is meant by "TEC data obtained from Precise Point Positioning". PPP – approach for determination of static and kinematic point positioning. I think the sentence can be rewritten.
12. Introduce TECU prior using it (or at the first mentioning).
13. After 65 – "block where the air was **ionized**"
14. Is there any quantitative analysis of ionospheric/atmospheric changes due to ionizations? Although such coupled processes may take place, it is not clear to what extent they are important and whether they can produce detectable changes in TEC to several units or not. I suggest considering clarifying this in the text if no references exist, or give a concluding remark at the end of the manuscript on the need for further quantifications of processes.
15. L85 – Please, reference the source of information on focal mechanism.
16. L110 – Please, consider writing for vTEC "free electrons along the line-of-sight between the center of the Earth and GNSS satellite" or similar. "Free electrons perpendicular to the earth" sounds not accurate.
17. L120 – you first mention that TEC can be calculated with at least two different frequencies. In the next sentence you write that TEC is obtained from single-frequency receivers. Please, consider rewriting these sentences to be more specific.
18. Please, indicate that Kp index below 4 is considered as quite conditions in this study.
19. In the previous revision, authors found it is not necessary to transform frequencies to periods in Figure 7. Although this would provide better understanding of numbers, I would then instead clarify where is an energy peak (what is a frequency or period). It is also not clear what is shown in Figure 7. Are these Power Spectral Density plots? Why the amplitude is in TECu?
20. L290 – "phonemes"

---

## Author Response (AR2)

[revised manuscript text omitted]

**Reviewer#1**

480 After the first revision, the manuscript was substantially improved, clarifying data processing methodology and providing additional insight on the validity of outcomes made. I believe that the manuscript has already a potential to be published, though I think some minor clarifications should be made, along with technical corrections.

- Thank you for your favorable comments, your time and consideration. We have revised the
485 manuscript very carefully and seriously by taking into consideration all of your comments and suggestions.

**Minor suggestions:**

1. Please, indicate which GNSS stations are used in CODE GIM maps. This question is related to the previous revision (Major comment 1). In Figure 1 of the "answers to reviewer" you showed comparison
490 of CODE GIM and IGS, but does CODE use the same stations as chosen from IGS or different ones? If these are the same stations, what is a reason to provide the analysis based on CODE GIM interpolated maps if RINEX data for the same stations are available and discussed in the article?

- In the article we used stations of the IGS network. We obtained the receiver DCBs of these stations by IONEX files from CODE. In other words, VTEC values of these stations are used in the
495 production of CODE GIMs. Yes, we could also calculate the epicenter TEC values with the help of the surrounding stations, but here we also demonstrate the accuracy of the calculated TEC values of IGS stations using the GIM TEC values. It should not be a problem to expand the results with a different data set.

Also, for Figure 1 in "answers to reviewer" I cannot understand how the BIASes were calculated.

500  - BIASes and RMSE values were calculated using the following formula:

$$bias = \langle TEC_{GNSS} - TEC_{GIM} \rangle$$

$$RMSE = \sqrt{\langle (TEC_{GNSS} - TEC_{GIM})^2 \rangle}$$

There are definitely some peaks close or even reaching 2 TECu, which is comparable with the amplitude of the detected anomaly. For example, the negative peak for difference plot for station ANKR reaches ~2
505 TECu at 11/04. Please, clarify these points in the final text.

- In order to explain this in the article, we have to add the Figure 1 (in the "answers to reviewer" file) to the article. We think just making such a statement without the Fig. 1 causes confusion. The negative discrepancy on 11/04 occurred between the two data sets. GIM represents a more global model, while GNSS represents a more local model. So it is normal for such differences to
510 occur between them.

2. Please, indicate what accuracy of vTEC (absolute value) you expect in your calculations and how it was estimated.

- The epicenter vTEC values were estimated using Eq. 8-10 at line153. Since interpolation points are close (GIM grids), these methods can achieve vTEC values with accuracy below 1 TECU.

Also, as I wrote it earlier, Forbes et al., 2020 and Mendillo et al. 2002 do not discuss that TEC cannot exceed 30%, as it is now stated at L245. You may want to add this clarification to the text.

- We arranged this part as follows (at line245).
- "*The ionosphere has a significant day-to-day variability due to thermospheric dynamics even though quiet space weather (Forbes et al., 2000). Here, we selected the ±30% limits for the day-to-day variability of the ionosphere.*"

For Figure 6, I would also suggest showing that 30% is consistent with no abnormal conditions for the whole time period as shown in Figure 4. You may consider merging Figure 6 with Figure 3 or 4.

- We combined Figures 5 and 6. Thus, we better uncovered the relationship of 30% limit to no-abnormal conditions.
- We also revised the sentence at line252 as "*We showed in the graph that the ± 30% limits of DTEC variation are **generally** consistent with the no-abnormal condition of the running median method based on M ± 2σ.*"

3. Why in Figure 4 I do not find the same strong negative anomalies 11/09-11/13 as in Figure 5? Also, some positive anomalies are shown for stations BSHM and ANKR at 11/07, but I can't find them in Figure 4. Generally saying, is there consistency between station analyses and CODE GIM maps? If not, what is a reason for inconsistencies and which data are better (this is some part related to equation 1 above)?

- First of all, thank you for your attention and detailed review. The difference you mentioned attracted our attention. We also identified the source of the problem in our running median software (coded in Matlab). When analyzing GPS-TEC values, we accidentally used a 10-day moving median instead of using a 15-day moving median (we analyzed GIM-TEC values with 15-day moving median). Therefore, GPS-TEC and GIM-TEC results differed. We intervened in the problem and updated the analysis in Figure 5 (both anomalies and DTEC values). We also updated some numerical values in the text according to the edited version of Figure 5.

**Technical suggestions:**

4. Please, consider another word in the first sentence of the abstract rather than "popular".

- "…popular approach…" changed as "… *one of the cutting edge issues* …"

5. 1st sentence of first paragraph – it is mentioned that the ionosphere is a dispersive layer. Dispersive for what? If you mean electromagnetic signals – please indicate, otherwise the sentence sounds incomplete.

- The first sentence was changed as "*The ionosphere is a three-dimensional dispersive atmosphere layer for electromagnetic signals traveling between space and earth.*"

6. Second sentence – what about ions? Please, consider rewriting this sentence.

- The third sentence was revised as *"… their components are divided into atoms, which are negative electrons and **positive ions**"*

550    7. 5th line – "to the Earth".

- Revised in first sentence of introduction section.

8. 5th line – I would write "To the first order, the degree of effect…."

- The sentence was revised as stated.

9. 6th line – "free electrons"?

555
- Electrons separated from molecules due to ionization. This is a very familiar phrase for ionosphere studies.

10. 8th line – please provide some references to daily, 27-day etc variations, I think that may provide reader better background.

- The below article was added as a reference for ionospheric variations.

560    "Vaishnav, R., Jacobi, C., Berdermann, J. (2019). Long-term trends in the ionospheric response to solar extreme-ultraviolet variations. In Annales Geophysicae, 37(6), 1141-1159."

11. Near 40 – Please clarify what is meant by "TEC data obtained from Precise Point Positioning". PPP – approach for determination of static and kinematic point positioning. I think the sentence can be rewritten.

565
- The sentence was revised as "*TEC data of Precise Point Positioning (PPP-TEC) calculating by PPP.PCF module in the Bernese software …*"
- The authors stated that they obtained PPP-TEC values in this way.

12. Introduce TECU prior using it (or at the first mentioning).

- We added to line43 "… 2-4 TECU (TEC unit = $10^{16}$el/m$^2$) …"

570    13. After 65 – "block where the air was ionized"

- Corrected.

14. Is there any quantitative analysis of ionospheric/atmospheric changes due to ionizations? Although such coupled processes may take place, it is not clear to what extent they are important and whether they can produce detectable changes in TEC to several units or not. I suggest considering clarifying this in

575    the text if no references exist, or give a concluding remark at the end of the manuscript on the need for further quantifications of processes.

- The quantitative value of anomalies is directly related to the method applied to TEC time series and selected limits (upper and lower bounds). We found here 1-2 TECU positive anomaly by adding ±2 standard deviations to the medians. This corresponds to the 95% confidence level. We

580    have already mentioned this in the conclusion section of the article (at line330). For example, if
       we set 1.5 standard deviation as a limit, these anomalies would probably be found as 4-5 TECU.
       There are previous studies where statistical analysis of abnormalities occurred in TEC time series
       before earthquakes. However, none of them focused on the quantitative value of anomalies. As I
       mentioned earlier, the quantitative value of anomalies is directly related to the selected limits
585    and is a relative value.

15. L85 – Please, reference the source of information on focal mechanism.

- The below article was added as a reference for information on focal mechanism.

"Wang, W., He, J., Hao, J., Yao, Z. (2018). Preliminary result for the rupture process of Nov. 13, 2017,
Mw7. 3 earthquake at Iran-Iraq border. Earth and Planetary Physics, 2(1), 82-83."

590    16. L110 – Please, consider writing for vTEC "free electrons along the line-ofsight between the center of
       the Earth and GNSS satellite" or similar. "Free electrons perpendicular to the earth" sounds not accurate.

- The sentence was changed as "This conversion provides the number of free electrons along the
  LOS between the center of the Earth and GNSS satellite."

17. L120 – you first mention that TEC can be calculated with at least two different frequencies. In the
595    next sentence you write that TEC is obtained from single-frequency receivers. Please, consider rewriting
       these sentences to be more specific.

- We revised the sentence as "*In recent years, some studies also showed that the TEC is calculated
  for single-frequency receivers by Precise Point Positioning (PPP) technique in which some
  parameters in the TEC calculation model are derived from IGS (Hein et al., 2016; Li et al., 2019).*"
600    - We also separated the next sentence, which describes how we achieved the TEC value in the
       study, as a new paragraph.

18. Please, indicate that Kp index below 4 is considered as quite conditions in this study.

- We added (Kp<4) in the last of sentence at line210.
- "Kp values indicate the disturbed magnetic condition between 7-11 November, whereas other
605    days have no magnetic activity values **(Kp < 4)**."

19. In the previous revision, authors found it is not necessary to transform frequencies to periods in
Figure 7. Although this would provide better understanding of numbers, I would then instead clarify
where is an energy peak (what is a frequency or period). It is also not clear what is shown in Figure 7. Are
these Power Spectral Density plots? Why the amplitude is in TECu?

610    - The Fourier frequencies are the output of short-time Fourier transform. The graphs in Fig. 7
       search for the TEC signal's predominant frequencies where their 'energies' reaches the peak
       level of amplitudes related to frequencies and time. They are not power spectral densities. The
       amplitudes show the TEC values per hertz.

- We have revised the sentence at line256 as "*The method provides the TEC signal's predominant frequencies where their 'energies' reaches the peak level of amplitudes related to frequencies and time. The amplitudes show the TEC values for per hertz.*"

20. L290 – "phonemes"

- Changed as "phenomena"

**Reviewer#2**

The authors have taken into account both reviewers' comments, resulting in a much improved manuscript.

- Thank you for your favorable comments, your time and consideration.

I only have one minor comment on the revised manuscript:

In my comments on the original manuscript, I raised questions about the calculation of the prompt penetration electric field shown in Figure 10. The authors have added a new subsection to explain the PPEF along with some references. Specifically, the authors state that "The penetration part of PPEFs (green line in Fig. 10) is calculated by the interplanetary data which is provided by the OMNI web site.". It is still not clear how exactly the calculation was done, from the solar wind V and B? I checked the reference Manoj and Maus, 2012 and realized that perhaps the penetration part of PPEF and the quiet part come from http://www.geomag.us/models/PPEFM/RealtimeEF.html? If yes, please acknowledge this website. If not, please clarify the method (present the equations/formulae and explain what are the observed quantities) used for calculating PPEF in this study.

- Yes, we obtained these values from the mentioned website and added a new sentence to the line303. "*The quiet and penetration part of PPEFs were obtained from http://www.geomag.us/models/PPEFM/RealtimeEF.html.*"